# TweedieMix: Improving Multi-Concept Fusion for Diffusion-based Image/Video Generation

**Gihyun Kwon**
KRAFTON
gkwon@krafton.com

**Jong Chul Ye**
Kim Jaechul Graduate School of AI, KAIST
jong.ye@kaist.ac.kr

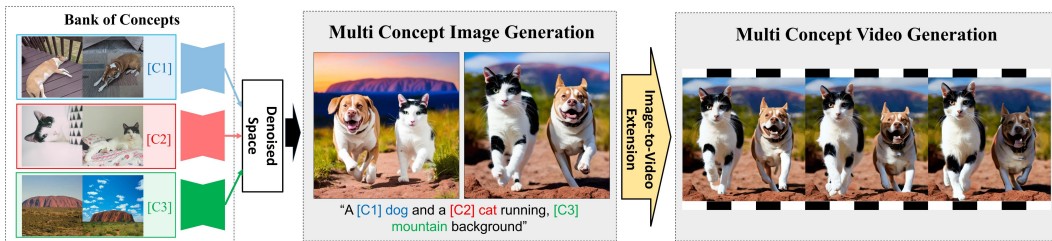

Figure 1: **Multi-concept Generation Results from TweedieMix.** Our model can generate high-quality multi-concept generation results on both of image and video domains. More results can be found in the experiment section.

## Abstract

Despite significant advancements in customizing text-to-image and video generation models, generating images and videos that effectively integrate multiple personalized concepts remains challenging. To address this, we present TweedieMix, a novel method for composing customized diffusion models during the inference phase. By analyzing the properties of reverse diffusion sampling, our approach divides the sampling process into two stages. During the initial steps, we apply a multiple object-aware sampling technique to ensure the inclusion of the desired target objects. In the later steps, we blend the appearances of the custom concepts in the de-noised image space using Tweedie's formula. Our results demonstrate that TweedieMix can generate multiple personalized concepts with higher fidelity than existing methods. Moreover, our framework can be effortlessly extended to image-to-video diffusion models by extending the residual layer's features across frames, enabling the generation of videos that feature multiple personalized concepts. Results and source code are in our project page.[1]

## 1 Introduction

In recent years, text-to-image generation models (Rombach et al., 2022; Saharia et al., 2022; Ramesh et al., 2022) have made remarkable strides, empowering creatives to produce high-quality images simply by crafting text prompts. This success has quickly expanded into other domains like video and 3D scene generation (Zhang et al., 2023a; Esser et al., 2023; Poole et al., 2022; Xu et al., 2023; Zhou et al., 2023; Liu et al., 2023a), achieving impressive results. Significant progress has been made in developing models that can customize images for specific subjects or visual concepts (Kumari et al., 2023; Gal et al., 2022; Ruiz et al., 2023; Tewel et al., 2023). These have enabled new possibilities for content creation, allowing users to leverage their own personalized characters.

Despite significant advancements in customizing these models for specific subjects or visual concepts, a major challenge persists: generating images that effectively combine multiple personalized concepts. Existing methods (Kumari et al., 2023; Tewel et al., 2023) allow for joint training of models on multiple concepts or merging customized models to create scenes featuring more than

---

[1]https://github.com/KwonGihyun/TweedieMix

one personalized element. However, these approaches often struggle with semantically related concepts—such as cats and dogs—and have difficulty scaling beyond three concepts. For instance, Mix-of-Show (Gu et al., 2023) attempted to tackle multi-concept generation using disentangled Low-Rank (LoRA) (Hu et al., 2021) weight merging and regional guidance during sampling. Yet, issues like concept blending remain due to the complexities involved in weight merging. To address these limitations, ConceptWeaver (Kwon et al., 2024) introduced a training-free method that combines multiple concepts during inference by splitting the generation process into multiple stages.

In this paper, we introduce an enhanced, tuning-free approach for composing customized text-to-image diffusion models during the inference stage. Unlike previous methods that require weight merging or additional inversion steps for multi-object generation, our technique utilizes only the reverse sampling steps and divides the process into two main stages. First, we conduct multi-object-aware sampling using text prompts that include multiple objects, introducing a novel resampling strategy to further improve generation quality. In the second stage, we integrate custom concept models through object-wise region guidance. To ensure stable and high-quality sampling, we combine each custom concept sample within the intermediate denoised image space calculated using Tweedie's formula. To expand the versatility of our method, we also propose a training-free strategy for extending these custom concept-aware images into the video domain.

Our experimental results demonstrate that our method can compose images featuring semantically related concepts without incorrectly blending their appearances. Moreover, our model seamlessly handles more than two concepts, overcoming a common limitation of baseline approaches. The images generated closely align with the semantic intent of the input prompts, achieving high CLIP scores. Finally, our video outputs outperform existing fine-tuning-based custom video generation methods, underscoring the effectiveness of our proposed framework.

## 2 RELATED WORK

Text-to-image (T2I) generation models have seen remarkable advancements over the years, evolving from early Generative Adversarial Network (GAN)-based models (Esser et al., 2021; Zhang et al., 2017) to the latest diffusion-based approaches (Saharia et al., 2022; Rombach et al., 2022; Yu et al., 2023; Ramesh et al., 2022). The development of diffusion models has opened up a range of applications, including text-guided image editing (Hertz et al., 2023; Couairon et al., 2023; Mokady et al., 2022), image translation (Kwon & Ye, 2023; Tumanyan et al., 2023), and style transfer (Zhang et al., 2023c). Recently, the success of T2I models has seamlessly extended to other modalities, such as 3D scene and asset generation (Poole et al., 2022; Xu et al., 2023), as well as video generation (Zhang et al., 2023a; Esser et al., 2023; Zhou et al., 2023; Bar-Tal et al., 2024). This expansion has spurred research into applications like Image-to-3D (Liu et al., 2023a; 2024) and Image-to-Video generation (Xing et al., 2023; Zhang et al., 2023b), along with editing capabilities for 3D scenes (Park et al., 2024; Chen et al., 2023) and videos (Jeong et al., 2023; Ceylan et al., 2023).

Building upon these advancements, there has been a growing interest in customizing T2I models using user-provided images or visual concepts. The pioneering work of Textual Inversion (Gal et al., 2022) focused on optimizing textual embeddings to represent custom concepts, enabling the generation of images that reflect these custom concepts. Subsequent studies have enhanced performance by developing extended textual embeddings (Voynov et al., 2023; Li et al., 2023) and fine-tuning model parameters (Kumari et al., 2023; Ruiz et al., 2023; Tewel et al., 2023), leading to more efficient and flexible customization options. Recently, framework of mutli-concept extraction (Hao et al., 2024) from single image has been proposed as an improved framework.

Extending beyond single-concept frameworks, researchers have explored methods for incorporating multiple concepts into customized models which use joint training to embed multiple concepts simultaneously or weight merging of single-concept customized model parameters (Kumari et al., 2023; Han et al., 2023; Tewel et al., 2023). However, these methods face challenges when scaling to a larger number of concepts or when dealing with semantically similar concepts, often resulting in the concept blending or disappearance of specific concepts. To address these issues, recent work like Mix-of-Show (Gu et al., 2023) applied regional guidance during the sampling process using merged weights to mitigate concept blending. Despite this improvement, the approach still requires additional optimization steps for weight merging.

As a further enhancement, ConceptWeaver (Kwon et al., 2024) introduced a training-free method that combines multiple concepts during the reverse sampling stage by dividing the inference process into multiple stages. While this method reduces the need for optimization, it suffers from longer inference times due to extra inversion steps and has limitations in flexibility, as it requires manipulation of attention layers. In contrast to previous methods, our approach eliminates the need for additional optimization or inversion steps and avoids direct manipulation of attention layers. This results in the production of higher-quality images while maintaining efficiency and flexibility in generating customized content featuring multiple concepts.

## 3 BACKGROUNDS

**Text-to-image Sampling with Classifier-free Guidance.** Denoising Diffusion Implicit Models (DDIMs) (Song et al., 2020) modify the reverse diffusion process of Denoising Diffusion Probabilistic Models (DDPMs) (Ho et al., 2020) to be deterministic and non-Markovian, allowing for more efficient sampling. As explained in the previous works (Kwon et al., 2023; Chung et al., 2023), the DDIM sampling process is divided into two parts: denoising with Tweedie's formula (Kim & Ye, 2021; Chung et al., 2022) and renoising part. If the score estimation model $\epsilon_\theta$ is trained the various text conditioned $\mathbf{c}$, we can obtain the text-guided DDIM update rule given by:

$$\boldsymbol{x}_{t-1} = \underbrace{\sqrt{\bar{\alpha}_{t-1}}\hat{\boldsymbol{x}}[\boldsymbol{\epsilon}_\theta(\boldsymbol{x}_t, t, \mathbf{c})]}_{\text{Denoising}} + \underbrace{\sqrt{1 - \bar{\alpha}_{t-1}}\boldsymbol{\epsilon}_\theta(\boldsymbol{x}_t, t, \mathbf{c})}_{\text{Re-noising}}, \tag{1}$$

$$\hat{\boldsymbol{x}}[\boldsymbol{\epsilon}_\theta(\boldsymbol{x}_t, t, \mathbf{c})] := (\boldsymbol{x}_t - \sqrt{1 - \bar{\alpha}_t}\boldsymbol{\epsilon}_\theta(\boldsymbol{x}_t, t, \mathbf{c}))/\sqrt{\bar{\alpha}_t}, \tag{2}$$

where $\boldsymbol{\epsilon}_\theta(\mathbf{x}_t, t, \mathbf{c})$ is the predicted noise at time step $t$ with text condition $\mathbf{c}$. Also, $\alpha_t = 1 - \beta_t$ and $\bar{\alpha}_t = \prod_{s=1}^{t} \alpha_s$ in which $\beta_t \in (0, 1)$ is a variance schedule.

In general text-to-image sampling, the model use Classifier-free guidance (CFG) (Ho & Salimans, 2022) which enhances conditional generation by combining the outputs of a conditional model and an unconditional model within the diffusion framework. This method allows for stronger conditioning without the need for an external classifier. During training, the model $\boldsymbol{\epsilon}_\theta(\mathbf{x}_t, t, \mathbf{c})$ is trained both with and without conditioning information $\mathbf{c}$ (e.g., text embeddings), where $\mathbf{c}$ is set to a null token with a certain probability $p_{\text{uncond}}$.

At inference time, the guided noise prediction and sampling update is computed as:

$$\boldsymbol{\epsilon}_{\mathbf{c}}^w(\mathbf{x}_t) = \boldsymbol{\epsilon}_\theta(\mathbf{x}_t, t, \varnothing) + w\left(\boldsymbol{\epsilon}_\theta(\mathbf{x}_t, t, \mathbf{c}) - \boldsymbol{\epsilon}_\theta(\mathbf{x}_t, t, \varnothing)\right), \tag{3}$$

$$\boldsymbol{x}_{t-1} = \sqrt{\bar{\alpha}_{t-1}}\hat{\boldsymbol{x}}[\boldsymbol{\epsilon}_{\mathbf{c}}^w(\boldsymbol{x}_t)] + \sqrt{1 - \bar{\alpha}_{t-1}}\boldsymbol{\epsilon}_{\mathbf{c}}^w(\boldsymbol{x}_t), \tag{4}$$

where $w > 1$ is the guidance scale, $\boldsymbol{\epsilon}_\theta(\mathbf{x}_t, t, \mathbf{c})$ is the conditional prediction, and $\boldsymbol{\epsilon}_\theta(\mathbf{x}_t, t, \varnothing)$ is the unconditional or null-text prediction.

**Improved Sampling with CFG++.** The traditional CFG framework employs a method where the CFG scale is set to $w > 1$, extrapolating between the unconditional score estimation output and the conditional score estimation. This approach causes off-manifold issues, such as abnormal saturation of the posterior mean calculated using Tweedie's formula during the early stages of sampling. These issues lead to a degradation in the text-image alignment quality of the final output. To address this problem, Chung et al. (2024) proposed a new framework that improves upon CFG. By using a smaller CFG scale, they interpolate between the conditional and unconditional scores, thereby alleviating the off-manifold problem of the posterior mean. The formulation of CFG++ is as follows:

$$\boldsymbol{\epsilon}_{\mathbf{c}}^\lambda(\mathbf{x}_t) = \boldsymbol{\epsilon}_\theta(\mathbf{x}_t, t, \varnothing) + \lambda\left(\boldsymbol{\epsilon}_\theta(\mathbf{x}_t, t, \mathbf{c}) - \boldsymbol{\epsilon}_\theta(\mathbf{x}_t, t, \varnothing)\right), \tag{5}$$

$$\boldsymbol{x}_{t-1} = \sqrt{\bar{\alpha}_{t-1}}\hat{\boldsymbol{x}}[\boldsymbol{\epsilon}_{\mathbf{c}}^\lambda(\boldsymbol{x}_t)] + \sqrt{1 - \bar{\alpha}_{t-1}}\boldsymbol{\epsilon}_\theta(\mathbf{x}_t, t, \varnothing), \tag{6}$$

where $0 < \lambda < 1$ is the guidance scale, $\boldsymbol{\epsilon}_\theta(\mathbf{x}_t, t, \mathbf{c})$ is the conditional prediction, and $\boldsymbol{\epsilon}_\theta(\mathbf{x}_t, t, \varnothing)$ is the unconditional or null-text prediction. The notable difference between CFG and CFG++ is that this framework use *unconditional* score in the renoising part. See Chung et al. (2024) for the mathematical motivation.

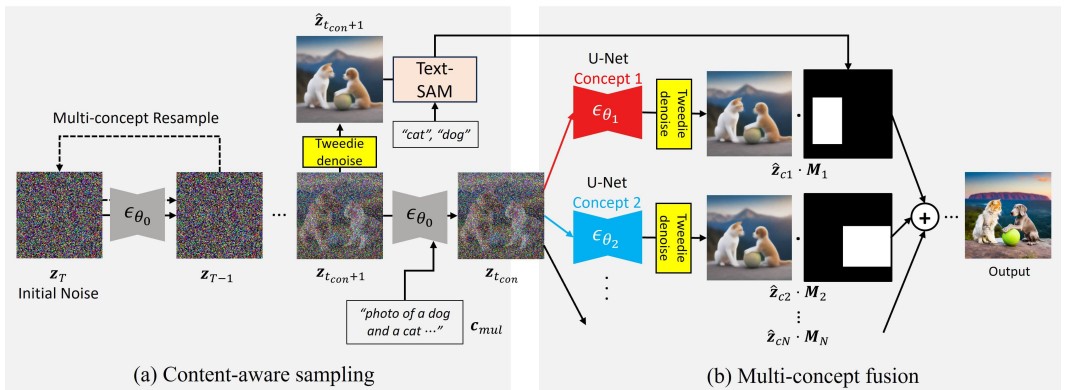

Figure 2: **Method Overview.** (a) To enhance the multi-object generation of text-to-image model, we use content-aware sampling in which we sample the image with non fine-tuned model $\epsilon_{\theta_0}$ and multi-object aware text $\mathbf{c}_{mul}$. In the intermediate step $t_{con}$, we extract mask from the images denoised with Tweedie's formula. (b) After $t_{con}$, we apply custom concept using region-wise guidance and concept-wise finetuned models. We propose to region-wise mixing of different models in Tweedie's denoised space.

# 4 METHODS

Building on the previous sampling strategy, we propose a methodology to combine individually fine-tuned U-Net models for custom concepts, applied exclusively during the sampling stage. In our approach, the fine-tuned U-Nets are models that have been adapted using various methods, such as weight fine-tuning as proposed in DreamBooth (Ruiz et al., 2023), Custom Diffusion (Kumari et al., 2023), or low-rank adaptation (Hu et al., 2021). This flexibility allows us to utilize different types of fine-tuned models. We refer to the base model that has not been fine-tuned as $\epsilon_{\theta_0}$, and the model fine-tuned for the $n_{th}$ concept as $\epsilon_{\theta_n}$.

**Content-aware Sampling.** An overview of our method is shown in the Figure 2. Our sampling method is divided into two stages. First, as illustrated in Figure 2(a), we perform sampling up to a specific timestep $t_{con}$ using the base model $\epsilon_{\theta_0}$ that has not been fine-tuned for any custom concepts. This approach addresses the issue where models fine-tuned on multiple concepts easily lose the ability to generate multiple objects. To prevent the loss of major basic components during the sampling process, we initially use the non fine-tuned model and a multiple context-aware text prompt $\mathbf{c}_{mul}$ that includes all the basic objects (e.g., *"a cat and a dog playing with a ball, mountain background"*) for sampling. In this stage, we utilize the CFG++ framework introduced earlier for sampling. This is crucial not only to achieve higher text-to-image alignment performance but also because it's important to use a smoothly varying posterior mean—not an off-manifold one—in the denoised image space where we will later perform multi-concept fusion. Also, we use latent diffusion framework in which the sample is conducted on latent space $\mathbf{z}$.

**Regional Mask Extraction.** In Figure 2(a), at the step just before $t_{con}$, we extract regional masks needed for the subsequent concept fusion sampling. In existing methods, this process was cumbersome as it involved extracting masks from pre-sampled images, performing inversion to convert them back into initial noise, or manually applying regional guidance. Our approach streamlines this process, allowing for more efficient and automated integration of regional masks during sampling.

More specifically, at timestep $t_{con} + 1$, we perform one-step denoising using Tweedie's formula and pass the result through the decoder to obtain an intermediate image $\mathcal{D}(\hat{\mathbf{z}}[\epsilon_{\mathbf{c}_{mul}}^{\lambda}(\mathbf{z}_{t_{con}+1})])$. This intermediate image, along with the words corresponding to each object (e.g. 'a dog', 'a cat'), is fed into a pre-trained off-the-shelf text-guided segmentation model to obtain region mask information for each object $M_1, M_2, \ldots$. At this point, instead of using precise dense masks, we extract and use the rectangular regions that enclose the respective masks. For the background, we set the mask as rest of object regions such as : $1 - \sum(M_i)$. Optionally, instead of directly segmenting the intermediate image at timestep $t_{con} + 1$, we can improve the segmentation quality by performing a few more additional sampling steps to $\hat{\mathbf{z}}_{t_{con}+1}$ and use them as the source for segmentation model.

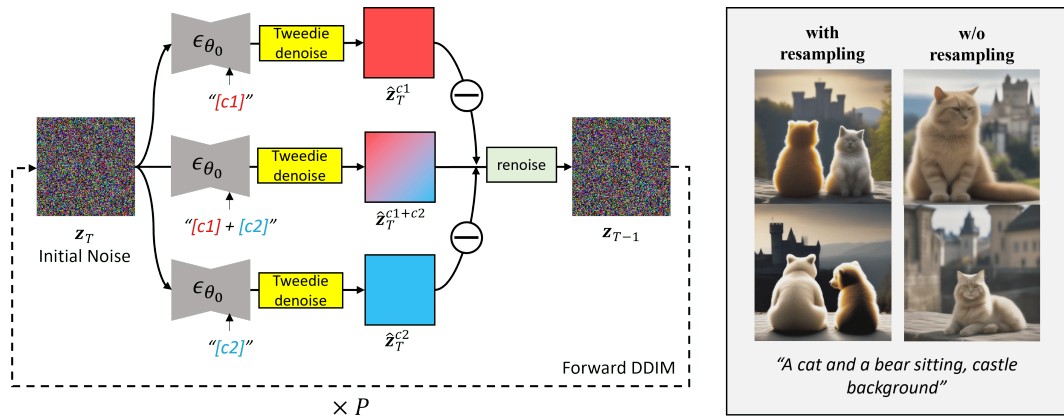

Figure 3: **Resampling Strategy.** To improve the multi-object sampling in content-aware sampling stage, we use resampling strategy. At initial timestep $T$, we subtract the single-concept samples from multi-concept samples to fortify the multi-concept text condition. This process is again calculated in the denoised space using Tweedie's formula. With the denoised image visualizations, we can see the effectiveness of our proposed resampling.

**Multi-concept Resample Strategy.** Despite the improved quality of using CFG++, we experimentally observed that it remains challenging to generate images containing multiple objects simultaneously due to the inherent limitations of the text-to-image (T2I) model performance. To address this issue, we discovered that it is possible to set an optimal starting point that enables the model to better generate multiple objects by performing multiple resampling steps at the initial phase, specifically at the most noisy timestep $T$, which significantly influences the overall quality of image generation.

Specifically, as illustrated in Figure 3, we sample the initial noise $z_T$ and perform denoising to proceed to the next timestep. At this point, we adjust the denoising output by subtracting the denoised samples from single-object text conditions $c_1, c_2, \ldots, c_n$ (e.g. *"a cat is playing with a ball"*, *"a dog is playing with a ball"*) from the denoised output obtained using the multiple-object text condition $c_{mul}$ (e.g. *"a cat and a dog is playing with a ball"*), which includes all the objects. Therefore, our sampling step at timestep $T$ can be described as:

$$\hat{z}_{\mathrm{adj}} = N\hat{z}[\boldsymbol{\epsilon}^{\lambda}_{\mathbf{c}_{\mathrm{mul}}}(z_T)] - \sum_{i=1}^{N} \hat{z}[\boldsymbol{\epsilon}^{\lambda}_{\mathbf{c}_n}(z_T)]. \tag{7}$$

$$z_{T-1} = \sqrt{\bar{\alpha}_{T-1}}\hat{z}_{\mathrm{adj}} + \sqrt{1 - \bar{\alpha}_{T-1}}\boldsymbol{\epsilon}_\theta(\mathbf{z}_T, T, \varnothing), \tag{8}$$

where $\boldsymbol{\epsilon}^{\lambda}_{\mathbf{c}}(z_T)$ represents the score output at timestep $T$ conditioned on text $c$. By moving to the next timestep $T-1$ using this adjusted denoised output, we observed that the multiple-object text condition $c_{mul}$ is more prominently emphasized in the generation process. After the mixing the denoised images, re-noising is performed using the unconditional score as suggested in CFG++. Subsequently, we perform DDIM forward sampling from $T-1$ back to the initial timestep $T$ using $c_{mul}$ and repeat this process $P$ times to amplify the effect. This resampling strategy effectively enhances the model's ability to generate images containing multiple objects by refining the initial starting point.

To clearly show the effect of our proposed resampling strategy, we show the Tweedie's denoised visualization output at timestep $t_{con}$ in the right side of Figure 3. When we sample the output without resampling, only one target object appears on the image, while the output from our proposed method shows multiple objects.

**Multi-concept Fusion Sampling.** After completing the previous content-aware sampling stage, we combine the custom concept-aware models in the subsequent steps. Since we have already obtained the sampling for multiple objects and the corresponding regional masks in the previous stage, we can apply the custom concepts to each different regions. Previous methods (Gu et al., 2023; Kwon et al., 2024; Yao et al., 2024) have also utilized region-wise sampling, but they mostly adopted approaches that mix concepts in the cross-attention features of the U-Net. While these methods

allow for differentiated sampling by region, applying mask regions can sometimes cause unwanted modifications due to the very small size of the bottleneck layer. Additionally, since all operations must be performed within a single model, there is a drawback of reduced flexibility due to limitations on the types of customized models that can be used.

To address these issues, we propose a method of mixing each concept in the denoised space using Tweedie's formula as shown in Figure 2(b). This approach enables more stable multi-concept fusion than using attention maps or noisy latent spaces. Since we leverage the framework of CFG++, we can confine the denoised space to exists on the same manifold. Therefore, it can correct the differences between outputs that occur when using different fine-tuned models. It also allows us to combine various models that have been fine-tuned in different ways for each custom concept (e.g., LoRA, key-value fine-tuning, etc.). Therefore, our multi-concept fusion sampling can be expressed as follows:

$$\boldsymbol{z}_{t-1} = \sqrt{\bar{\alpha}_{t-1}}\{\sum_{i=1}^{N} M_i \cdot \hat{\boldsymbol{z}}[\tilde{\boldsymbol{\epsilon}}_i]\} + \sqrt{1 - \bar{\alpha}_{t-1}}\boldsymbol{\epsilon}_{\theta_0}(\boldsymbol{z}_t, t, \varnothing), \tag{9}$$

$$\tilde{\boldsymbol{\epsilon}}_i = \boldsymbol{\epsilon}_{\theta_0}(\boldsymbol{z}_t, t, \varnothing) + \lambda\left(\boldsymbol{\epsilon}_{\theta_i}(\boldsymbol{z}_t, t, \mathbf{c}_i) - \boldsymbol{\epsilon}_{\theta_0}(\boldsymbol{z}_t, t, \varnothing)\right), \tag{10}$$

where $\boldsymbol{\epsilon}_{\theta_i}(\boldsymbol{z}_t, t, \mathbf{c}_i)$ is estimated score output from U-net fine-tuned on $i_{th}$ custom concept using single-concept aware text prompt $\mathbf{c}_i$ in which the sentence exclusively contains word corresponding to $i_{th}$ concept. (e.g. $\mathbf{c}_1$ : *"A [c1] cat playing with a ball."* , $\mathbf{c}_2$ :*"A [c2] dog playing with a ball."*, ...). In the unconditional score estimation using null-text prompts, we found that using the null-text outputs from each concept's fine-tuned model $\boldsymbol{\epsilon}_{\theta_i}$ hinders the natural mixing between objects and background. To address this issue, we commonly used the unconditional score estimation output obtained from the non fine-tuned model $\boldsymbol{\epsilon}_{\theta_0}$, across all concepts.

**Extension to Video Domain.** Using the proposed sampling strategy, we successfully generated images that accurately represent multiple custom concepts. Building on this success, we propose a novel method to extend multi-concept generation into the video domain.

Previous methods (Molad et al., 2023; Wei et al., 2023) for video generation of custom concepts have primarily focused on single concepts, often employing techniques that fine-tune the attention layers of text-to-video models. These approaches require extensive computational resources and significant training time due to the need for video model fine-tuning. The

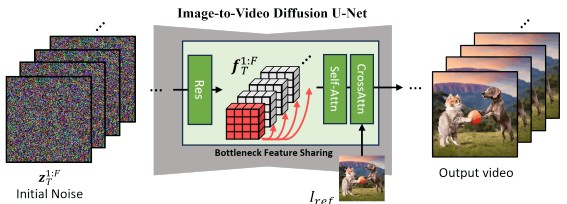

Figure 4: **Method for Video Extension.** To preserve the context of reference image which is generated from our multi-concept sampling strategy, we propose to inject the residual features of first frame to the other frame features.

problem becomes more serious when dealing with multiple custom concepts. Additionally, these methods tend to suffer from overfitting, leading to a substantial degradation in performance when generating multiple objects.

To overcome these limitations, we introduce a simple yet powerful training-free strategy. Since we already have well-generated images of multiple custom concepts from earlier steps, our approach focuses on animating these images using a pre-trained image-to-video model (Zhang et al., 2023b) to enable multi-concept video generation. However, merely conditioning the video generation on these images often causes the generated frames to lose the context of the original concepts easily. Our goal is to ensure that the video consistently maintains the appearance of the same custom concepts from start to finish.

To address this issue, we drew inspiration from feature injection methods recently used in 2D image editing (Tumanyan et al., 2023). In Figure 4, we begin by sampling with the image-to-video model using random noise $\boldsymbol{z}_T^{1:F}$ and image conditions $I_{ref}$ which is sampled from our method. We observed that the initial timestep $T$ plays a critical role in determining the semantics of subsequent frames. During the first timestep $T$ when utilizing the I2V U-Net, we transfer the U-Net features from the first frame (Reference Frame) to the following frames.

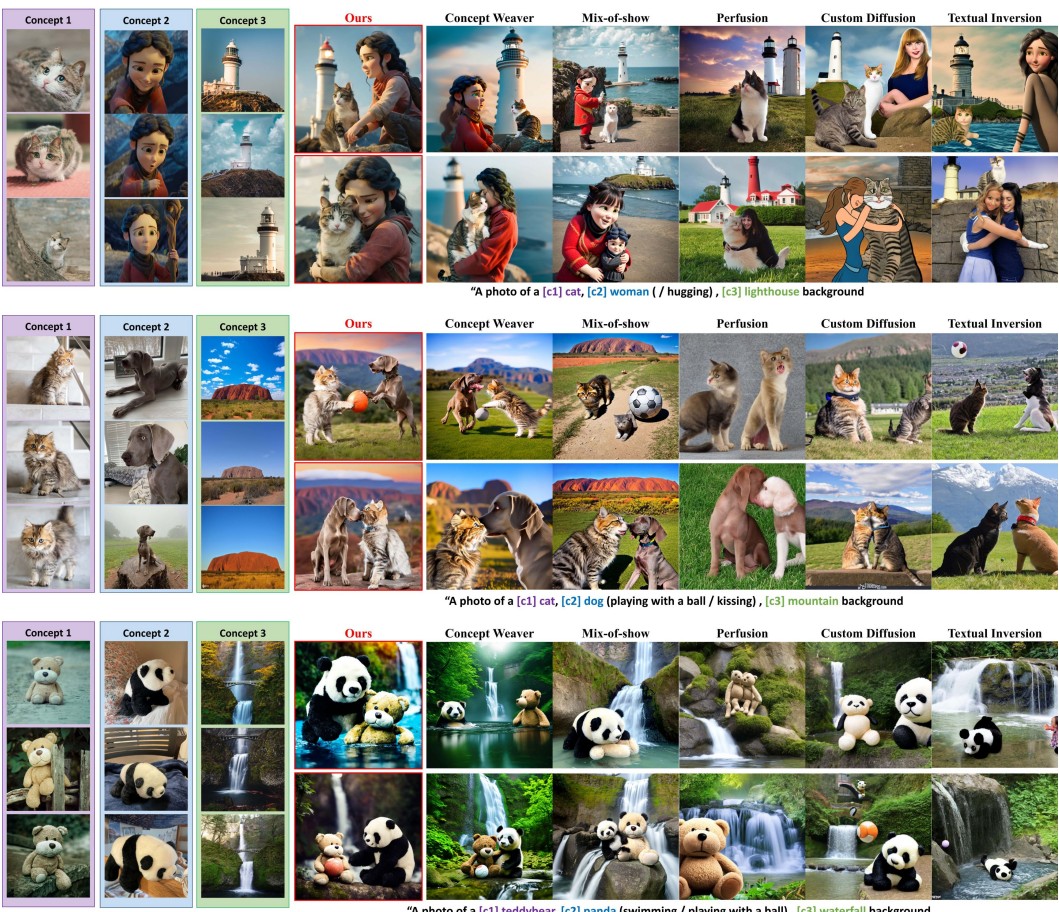

Figure 5: **Qualitative Evaluation of Multi-Concept Image Generation.** We evaluate the image generation quality of our method in comparison to baseline approaches, using prompts that incorporate each concept displayed on the left. In the overall results, our method maintains the appearance of the target concepts without any concept missing problems, whereas the baseline methods fails to preserve the identity of the concepts or generate the intended action corresponding the text.

Transferring features from all layers, however, led to artifacts or a complete loss of motion. To mitigate this, we carefully selected which layers to inject features into. We found that injecting features from the residual layers in the bottleneck of the U-Net allows us to preserve the appearance while still generating the desired motion in the video.

Moreover, we noticed that simply replacing the features of the first frame significantly reduced motion. To prevent this, we employ linear interpolation between the features of the first frame and those of the other frames. Denoting the residual features of I2V U-Net as $\boldsymbol{f}^{1:F}$ if the total frame number is $F$, the intermediate features at timestep $T$ can be expressed as follows:

$$\boldsymbol{f}_T^{2:F} \leftarrow \eta \boldsymbol{f}_T^1 + (1 - \eta) \boldsymbol{f}_T^{2:F} \tag{11}$$

For feature interpolation, we used all the lowest resolution U-Net bottleneck blocks (Midblocks) and the first residual layer of upsampling block. For the lowest resolution blocks, we set $\eta = 1$, and for the first upsampling block, we set $\eta = 0.3$.

## 5 EXPERIMENTAL RESULTS

### 5.1 EXPERIMENTAL SETTINGS

**Baseline Methods.** We compare the proposed method with various existing concept personalization techniques. For this, we adopted earlier methods such as Textual Inversion (Gal et al., 2022), Custom

| Method | CLIP score | | DINO score | User Study | | |
|---|---|---|---|---|---|---|
| | Text sim↑ | Image sim↑ | Image sim↑ | Text match↑ | Concept Match↑ | Realism↑ |
| Textual Inversion | 0.3472 | 0.7692 | 0.4420 | 3.01 | 2.05 | 2.16 |
| Custom Diffusion | 0.3343 | 0.7456 | 0.4965 | 2.82 | 2.63 | 2.48 |
| Perfusion | 0.3222 | 0.7182 | 0.4201 | 2.01 | 2.44 | 1.88 |
| Mix-of-show | 0.3581 | 0.7839 | 0.5211 | 3.82 | 3.77 | 3.56 |
| Concept Weaver | 0.3707 | 0.8095 | 0.5352 | 4.14 | 4.32 | 4.11 |
| ß **TweedieMix (ours)** | **0.3816** | **0.8311** | **0.5950** | **4.56** | **4.71** | **4.47** |

Table 1: **Quantitative Evaluation of Multi-Concept Image Generation.** Our model outperforms baselines in overall scores for both of text-alignment and image-alignment CLIP scores. Also our model shows improved perceptual quality with obtaining the highest scores in user preference study.

Diffusion (Kumari et al., 2023), and Perfusion (Tewel et al., 2023), and show the results from the models joint-trained with multi-concept data. Additionally, we used Mix-of-Show (Gu et al., 2023), a state-of-the-art method for multi-concept generation. We conducted experiments using the official source code of the model. Since this model requires manual region guidance, we provided region guidance for each image. Furthermore, we incorporated the more advanced ConceptWeaver (Kwon et al., 2024) method as the latest baseline for comparison.

**Evaluation Settings.** For the evaluation dataset, we utilized the dataset proposed in the prior work, drawing from various data sources for both quantitative and qualitative analyses. For the quantitative evaluation, we selected 32 distinct concepts from the Custom Concept 101 dataset (Kumari et al., 2023), organized into 10 unique combinations. These concepts span a wide variety of categories, including animals, humans, natural scenes, and objects. For the qualitative analysis, we expanded the concept pool by adding three animated character concepts sourced from YouTube Blender Open Movie. All the dataset contains 5 - 8 images per each concept.

For evaluation metrics, we also borrowed the protocol provided in the recent work (Kwon et al., 2024). We evaluate our method against baseline approaches by measuring Text-alignment (Text-sim) and Image-alignment (Image-sim) using CLIP scores (Radford et al., 2021). Text-alignment calculates the cosine similarity between the CLIP embedding of the generated image and the CLIP embedding of the text prompt. To better assess our model's ability to generate multiple concepts, we use modified the standard Image-alignment metric. This adaptation computes cosine similarity between the visual embeddings of specific concept regions and the embeddings of the corresponding target concepts. We calculate these metrics across 100 unique images generated by each model. The evaluation uses ten combinations of multiple concepts, with each combination containing more than three concepts. For more thorough evaluation of our concept generation performance, we calculated same image similarity score on pre-trained DINOv2 (Oquab et al., 2024) model. We report the average Text-alignment and Image-alignment scores across all generated images.

## 5.2 MULTI-CONCEPT IMAGE GENERATION RESULTS

**Qualitative Evaluation.** To compare our method with the baselines, we generated images combining three different custom concepts. Our results included both simple text prompts and more complex prompts where objects interact with each other. Figure 5 presents our qualitative evaluation results. In the case of early models like Textual Inversion, Perfusion, and Custom Diffusion, we observed that due to training failures caused by joint training, the target concepts did not appear accurately in the images, and the generated images did not match the text prompts. Even with the multi-concept sampling model, Mix-of-Show, although multiple custom objects were generated, their appearance did not accurately follow the target concepts, and some results did not align with the target text. When compared to the latest model, ConceptWeaver, this model reflected custom concepts and text conditions to a certain extent, but there were slight differences in the texture or color appearance of the images from the target, and some images showed unnecessary modifications. Our results more accurately represented the target concepts than the baselines and more naturally incorporated the text conditions into the generated images.

**Quantitative Evaluation.** Table 1 shows the comparison results of CLIP, DINO scores and human preference user study between our model and baseline models. In terms of CLIP and DINO scores, the early models demonstrated relatively lower values for both text-image alignment scores

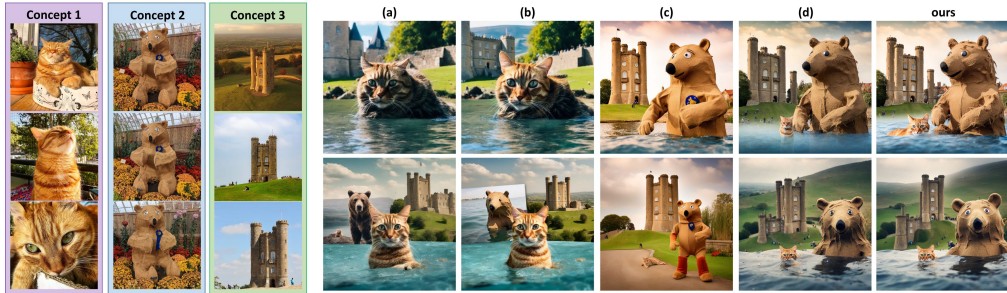

"A photo of a [c1] cat, [c2] bear swimming, [c3] castle background

Figure 6: **Ablation Study on Image Generation.** To evaluate the proposed method components, we conduct ablation study. (a) Results without CFG++. (b) Results without using resampling strategy. (c) Results without content aware sampling. (d) Results with mixing in noisy latent space.

and image-image matching scores. While the scores of the latest multi-concept models showed significant improvement over the early models, our model achieved the best quantitative scores.

To validate the perceptual quality of our results, we conducted a user study with 20 different participants. In the user study, we followed the protocol of previous work. Specifically, we asked users three questions across different categories: 1) alignment with the text prompt(text-match), 2) whether all target concepts were accurately generated(concept-match), and 3) whether the generated image appeared realistic(realism). Each participant viewed 60 images and rated them on a 5-point scale, where 1 indicated "strongly disagree" and 5 indicated "strongly agree." The results of the user study clearly demonstrated that our model significantly outperformed the baseline models in terms of perceptual quality.

**Ablation Study.** We conducted an ablation study to evaluate the components of our proposed method in Figure 6. (a) First, we observed that some concepts disappeared when the resampling strategy was not used. (b) Next, we replaced the CFG++ framework with standard CFG. Without using CFG++, we found that the proposed resampling strategy did not function properly, so this setup excluded both CFG++ and resampling. In this case, we observed a decline in output quality, along with the disappearance of some concepts. (c) To demonstrate the effectiveness of the

|                         | CLIP score  |             |
| ----------------------- | ----------- | ----------- |
| **Settings**            | Text sim↑   | Image sim↑  |
| (a) w/o resample        | 0.3551      | 0.8078      |
| (b) w/o CFG++           | 0.3496      | 0.8020      |
| (c) w/o content sampling | 0.3427     | 0.8026      |
| (d) with eps mixing     | 0.3821      | 0.8146      |
| ours                    | 0.3872      | 0.8202      |

Table 2: **Ablation Study on Image Generation.** Quantitative evaluation on ablation study. Our best setting shows the highest scores.

proposed two-step strategy, we conducted an experiment without using our content-aware sampling in the initial steps and instead applied manual masks for fusion sampling from the start. In this case, we saw that some specific concepts disappeared, and the overall quality deteriorated. (d) Finally, to verify the effectiveness of the proposed method for mixing in the Tweedie's denoised space, we experimented with mixing concepts in the noisy latent space. While multi-concept images were generated successfully, we noticed a slight reduction in realism when generating small objects. When using our best setting, we achieved the highest image quality, and the multi-concept objects were accurately represented. This is further confirmed in our quantitative comparison in Table 2.

## 5.3 MULTI-CONCEPT VIDEO GENERATION RESULTS

**Qualitative Evaluation.** To evaluate our video generation model, we used existing video customization models as a baseline. Specifically, we employed the latest video customization model Dreamvideo (Wei et al., 2023). As it is originally designed for single-concept customization, we used jointly fine-tuned model using our multi-concept dataset.

Figure 7 presents the results of our qualitative comparison. In the case of the baseline Dreamvideo, it failed to generate multiple concepts, and even the generated single concept did not accurately represent the target concept. Although our method extends the Image-to-Video model, making direct

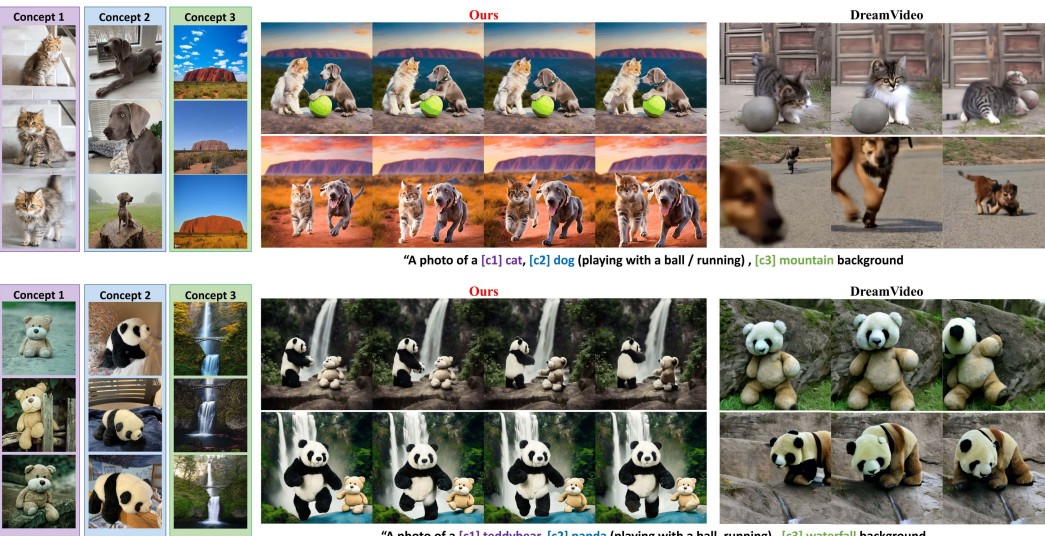

Figure 7: **Qualitative Evaluation of Video Generation.** Our model can generate high-quality multi-concept video generation while the baseline Dreamvideo fails to generate concept-aware videos.

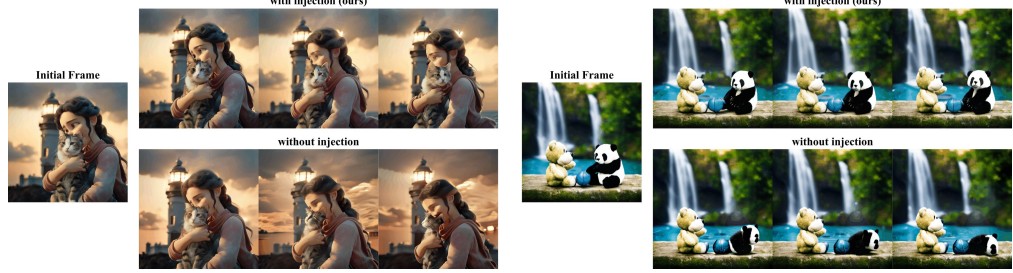

Figure 8: **Ablation study on Video Generation** Without using our proposed feature injection method, the output video shows degraded performance without preserving the inter-frame context.

comparisons less straightforward, our results demonstrate that multiple concepts were generated without mixing, and the continuity and context of consecutive frames were accurately preserved. We also conducted a user study to compare perceptual quality, where 20 participants were shown 4 videos, and their overall preference scores were recorded. In this study, our model scored **4.5** whereas the baseline DreamVideo scored 1.7, which further indicate the superiority or our results.

**Ablation Study.** Additionally, to assess the effectiveness of our proposed feature injection method, we compared the results with and without feature injection in Figure 8. When feature injection was not properly applied, we observed that the shape of the concept object was excessively distorted or disappeared entirely, resulting in a complete mismatch with the original context. On the other hand, when injection was used, the object's appearance from the first frame was consistently maintained throughout the video. This confirms that our method is actually effective in context preservation.

# 6   CONCLUSION

In this paper, we proposed a novel framework for generating multiple custom concepts. Our approach splits the sampling process into two main steps. In the initial phase, we perform multi-object sampling, introducing a resampling strategy to enhance performance. Following this, we extract masks in the intermediate steps and apply custom concepts to specific regions using these masks for regional sampling. By integrating the region-wise concept mixing within the denoised image space, we achieved superior results. Additionally, we extended our framework to the video domain through feature injection, enabling the generation of multi-concept videos. Experimental results demonstrate that our model outperforms baseline approaches, confirming improved generative performance.

**Acknowledgements** This work was supported by the National Research Foundation of Korea under Grant RS-2024-00336454 and by the Institute for Information & Communications Technology Planning & Evaluation (IITP) grant funded by the Korea government (MSIT) (RS-2019-11190075, Artificial Intelligence Graduate School Program, KAIST).

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

## A  MORE EXPERIMENTAL DETAILS

**Implementation Details.** We can utilize any customization framework to train a model fine-tuned for individual custom concepts to build the concept bank. Among these, we used major two methods: fine-tuning the cross-attention layer's key-value weights as proposed in Custom Diffusion and the DreamBooth method combined with low-rank adaptation. Half of the custom concept models we used were trained with low-rank adaptation, while the other half were trained using the Custom Diffusion framework.

As the backbone diffusion model, we used Stable Diffusion 2.1 or higher. In this paper, we primarily used the SDXL model as the base model, and for fair comparison, the baseline models were retrained using SDXL. However, for the latest models like Mix-of-Show and Concept Weaver, we found that adapting them to SDXL is not appropriate according to their proposed methods, so we used the official source code and settings provided in their respective papers for our experiments.

Regarding sampling hyperparameters, we set the reference timestep $t_{con}$ for content-aware sampling to $0.8T$, and we found that values between $0.8T$ and $0.7T$ did not significantly affect output quality. The total timestep is set to $T$=50, and the we used image resolution of 768x768. In terms of sampling time, it takes approximately 30 seconds using a single NVIDIA RTX 3090 GPU. For resampling, we used $P = 10$, and found no significant quality difference when using between 5 and 10 resampling steps.

For the text-guided segmentation model, we used the langsam (Medeiros, 2023) package, which combines Grounding DINO (Liu et al., 2023b) and Segment-Anything models (Kirillov et al., 2023), to obtain masks based on the text condition. Instead of dense masks, we used rectangular regions containing the mask, and in cases where masks for two concepts overlapped, we modified only the overlapping area to retain the shape of the original dense mask. If mask extraction failed, causing complete overlap between the two objects' masks or no mask was found, we did not proceed with additional sampling. To ensure non-overlapping masks for each concept, once the region for the first object was extracted, that area was excluded from the region range for the next object's extraction.

For the video model, we used the recently proposed image-to-video model, I2VGen-XL (Zhang et al., 2023b). For video sampling, we set $T$=50. The total number of frames was 16, with a resolution of 512x512. As described earlier, we performed feature injection at three residual layers (midblock 2, upsampling 1). This process took approximately 50 seconds on a single RTX 3090 GPU.

**Evaluation Details.** To calculate the image-alignment score, we followed the procedure outlined in the previous work, ConceptWeaver (Kwon et al., 2024). Since our generated images include multiple concepts, we couldn't rely on whole image similarity scores. Instead, we used a text-guided segmentation model to extract concept-specific images. For example, when evaluating images with '[c1] dog' and '[c2] cat,' we applied the segmentation model using the prompts 'dog' and 'cat' to obtain segmented masks. We then cropped the regions containing these masks and calculated the cosine similarity between the image embeddings of the extracted sections and the corresponding concept (training) images. Since baseline methods often fail to generate all concepts, we excluded scores for images lacking all foreground objects to ensure a fair comparison. For ablation study, we choose 15 different concepts from customconcept101 dataset and experimented with unique 5 different combinations. For CLIP score calculation, we also followed same protocol of quantitative evaluation. In DINO score calculation, we used the pre-trained DINO-v2-base model and reported cosine similarity between the embedding of target concept images and extracted generated object images.

For human preference evaluation, we gathered feedback from 20 participants aged 20-49. We created a survey set containing 10 generated images per baseline model and 10 questions. The models used for generating outputs were Textual Inversion, Custom Diffusion, Perfusion, Mix-of-Show, ConceptWeaver, and ours, resulting in a total of 60 generated images per survey set.

## B  EFFECT OF HYPERPARAMETERS

In order to show the effect of hyperparameter change, we additionally conducted experiment by varying the key hyperparameter components: timestep for content-aware sampling $t_{con}$ and the

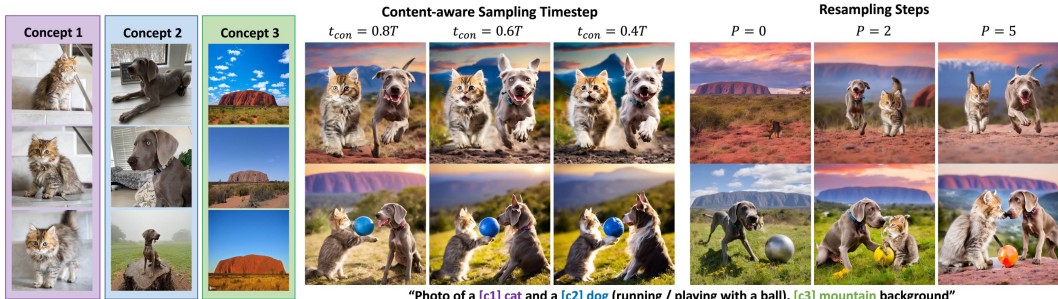

Figure 9: **Results with various hyperparameter settings.** With changing the content-aware sampling timestep, we can control how much each object contains the appearance of target custom concepts. With different number of resampling steps, we can observe the different ability of multi-object generation.

number of resampling steps $P$. In Figure 9, we show the generated results with different settings. With varying the content-aware sampling, we can obtain more custom-concept aware outputs with using larger timestep of $t_{con}$, and the output shows more generic concepts with using smaller $t_{con}$. The result means that if we use smaller steps for custom-concept fusion, the output becomes far from target custom concepts.

With changing the resampling steps, we can observe that there is significant concept missing problem when we do not use any resampling. With using small resampling steps, we can alleviate the concept missing partially, but still there exists some of the artifacts in the generated objects. With using more resampling steps, the multi-concept outputs are more clearly generated compared to other settings.

## C   TIME EFFICIENCY COMPARISON

To compare the time consumption between our proposed method and baselines, we calculated times taken for inference on single 768x768 image. For our method, single sampling path takes 28 seconds on single RTX3090 GPU, which includes time for segmentation. For our baseline Concept Weaver, the method takes about 65 seconds, which include template generation, inversion, segmentation and sampling. For our early baseline of Mix-of-show, the method takes about 23 seconds. Although the method requires additional weight merging stage for each concept combination, we did not considered it for fair comparison. For earlier baseline of Perfusion, Custom Diffusion, Textual Inversion takes about 10 seconds for sampling, which is almost same as vanilla sampling case. Considering the superior quality of our shown results, our proposed TweedieMix shows the best efficiency on multi-concept generation.

## D   RESULTS ON STYLE COMPOSITION

To verify that our proposed framework is not limited on specific custom concepts, we experimented with showing composition on various custom styles. In Figure 10, we show the output of various objects with different styles. When we see the output from vanilla sampling case, the generated outputs show difficulty in discriminating the style of each components. In our proposed method, the generated images contain separated objects with accurately reflecting the indicated style representations. The results shows that our method is not limited on specific custom concepts, but it can be extended to wider domain of image styles.

## E   ADDITIONAL RESULTS

**Results on more concepts.** To show the generation performance when using more than 3 concepts, we show the comparison results using 4 different concepts in Fig. 11. The results show that our proposed method can generate realistic multi-concept aware outputs even with fusing 4 different

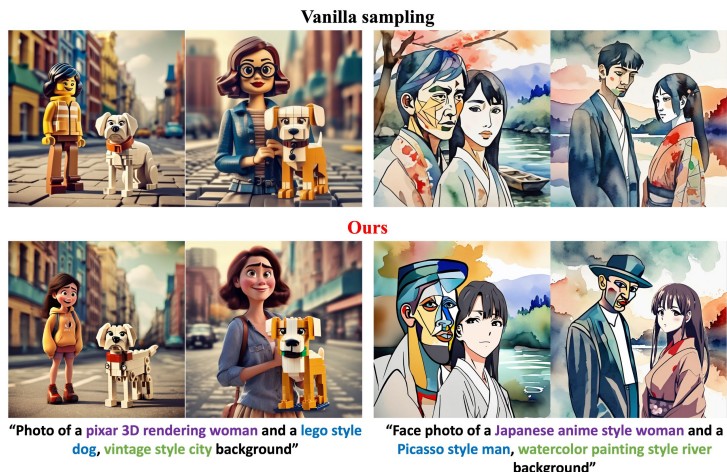

Figure 10: **Results with various style composition.** Our proposed method successfully combines multiple objects with different styles, while vanilla sampling method suffer from mixed styles between the objects.

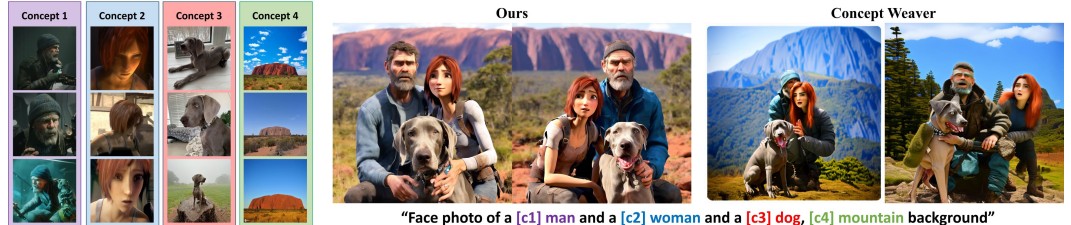

Figure 11: **Results on more concepts.** Our model can generate high-quality multi-concept generation results when using 4 custom concepts, while our baseline Concept weaver fails to reflect the appearance of target concepts.

concepts. Our baseline of Concept Weaver also can generate multiple objects, but the generated concepts often deviate from target concepts, and also suffer from severe artifacts.

**Detailed comparison with baseline.** In order to thoroughly compare the generation quality of our proposed method and baseline of Concept Weaver, we show the multiple generated images in Figs. 14,15,16. The results from our method successfully reflect the appearance of multiple custom concepts. In our baseline Concept Weaver, although the images contain the multiple objects, most of the generated custom concepts suffer from severe artifacts or unwanted modifications, and the generated outputs are unrealistic.

**Extension to Stable Video Diffusion.** To verify the flexibility of our video extension method, we applied same feature injection method on Stable Video Diffusion I2V in Figure 12. Our method can still prevent the excessive modification of target objects and generate high-quality multi-concept video.

In order to further show the detailed results, we present more results in Figure 17. The results show that our method can generate detailed multiple custom concepts while maintaining the semantic of given text prompts.

We also show the additional video results in Figure 18. The results again shows that our video generation method successfully synthesize multiple custom concepts while preventing extreme changes between the frames. For better visualization, we attached the video samples in our supplementary materials.

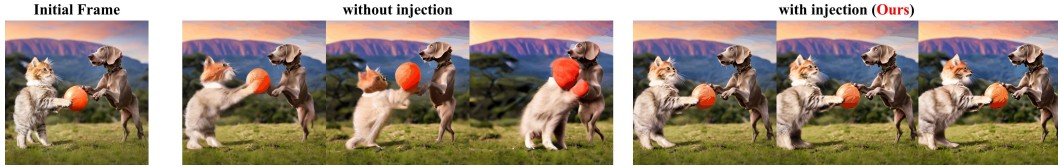

Figure 12: **Results from Stable Video Diffusion.** Our video extension can be easily adapted to other Image-to-video framework.

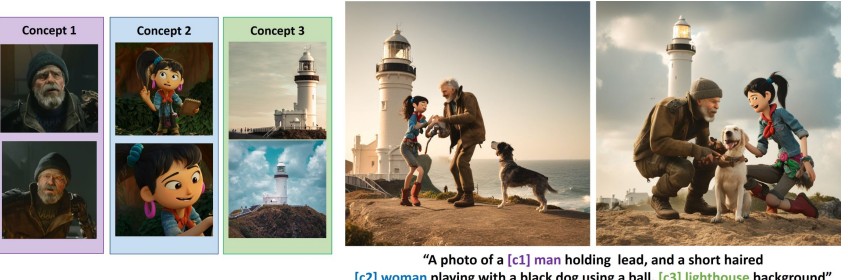

"A photo of a [c1] man holding lead, and a short haired
[c2] woman playing with a black dog using a ball, [c3] lighthouse background"

Figure 13: **Failure cases.** If the text condition becomes extremely complex, then the generated concepts loses the detailed appearance.

# F  LIMITATIONS

While our method demonstrates strong performance in multi-concept generation, it still has some limitations. When presented with highly complex or unrealistic text prompts, the performance in text alignment remains restricted. As shown in Figure. 13, we can observe that the method can generate multiple objects in the text conditions, but the objects loses its fine details of the target concept images, and the detailed text descriptions are slightly unaligned with the text. This issue stems from the constraints of the pre-trained Stable Diffusion model. We anticipate that using enhanced diffusion model backbones will help address this challenge in the future.

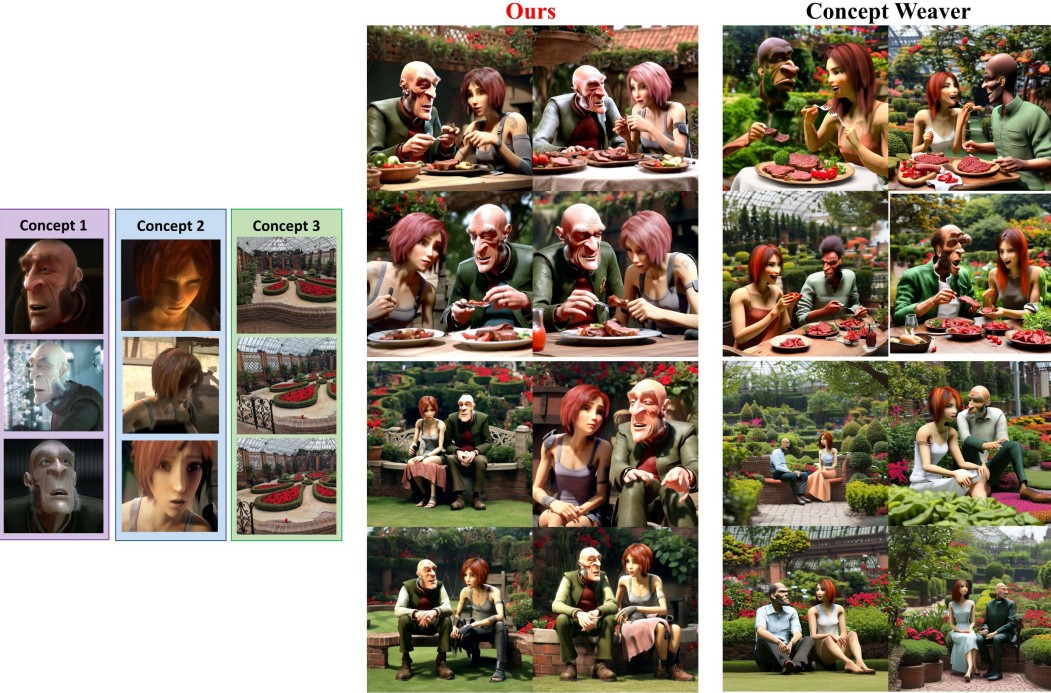

Figure 14: **Additional Comparison Results.** Our model can generate high-quality multi-concept generation results on image domains.

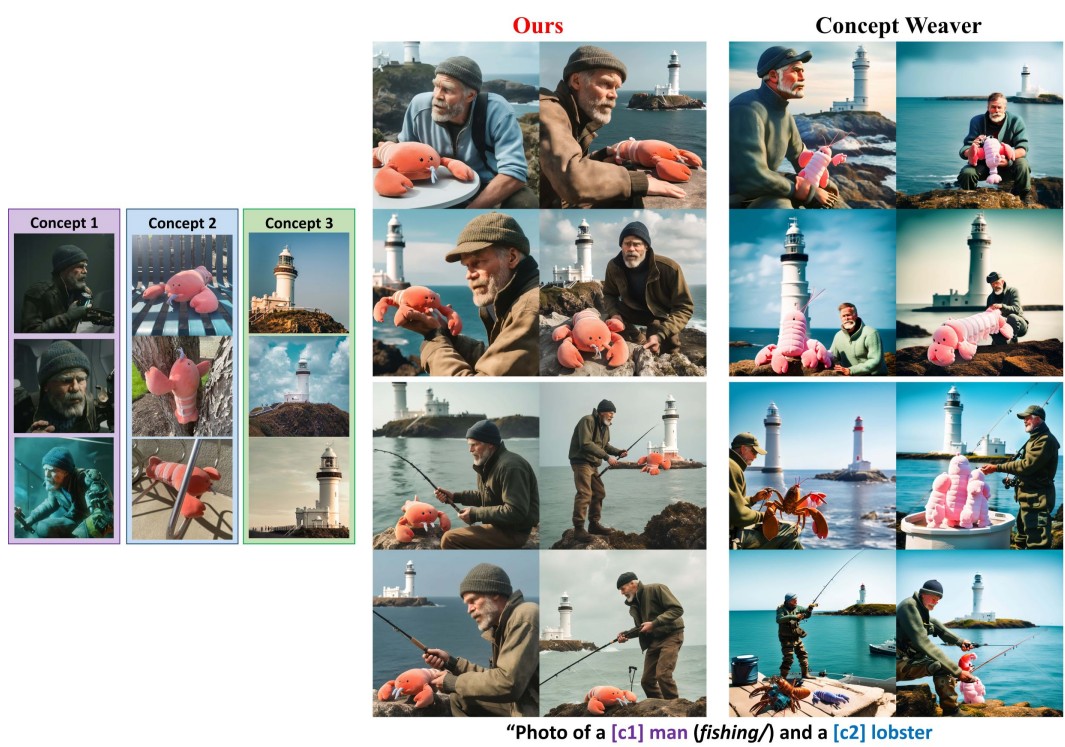

Figure 15: **Additional Comparison Results.** Our model can generate high-quality multi-concept generation results compared to Concept Weaver.

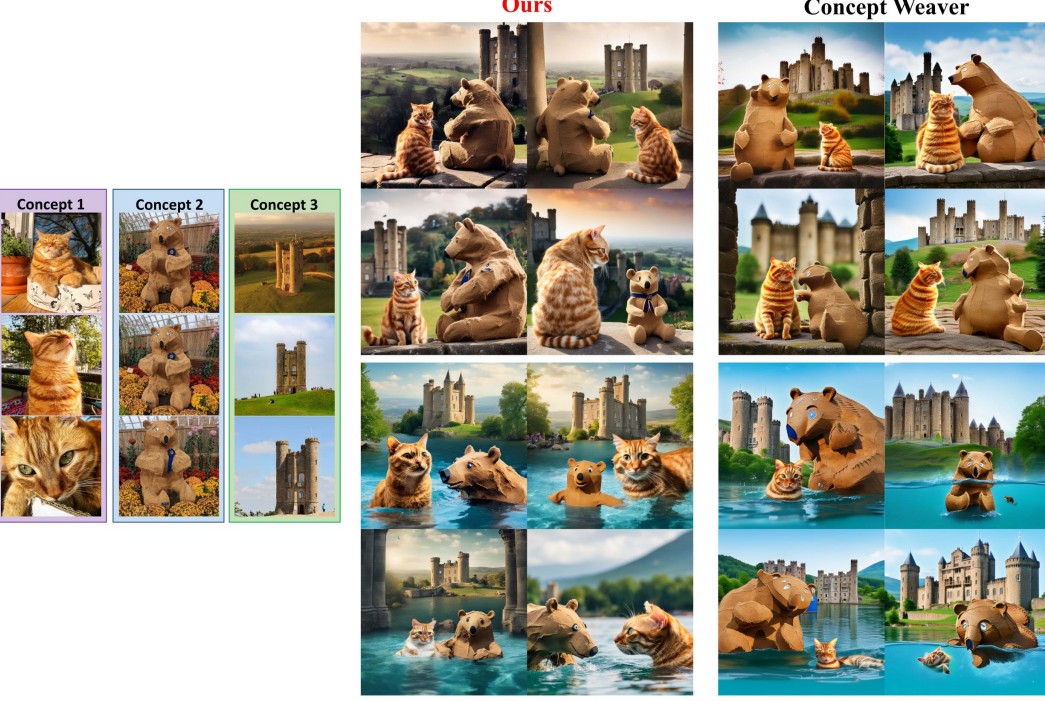

Figure 16: **Additional Comparison Results.** Our model can generate high-quality multi-concept generation results compared to Concept Weaver.

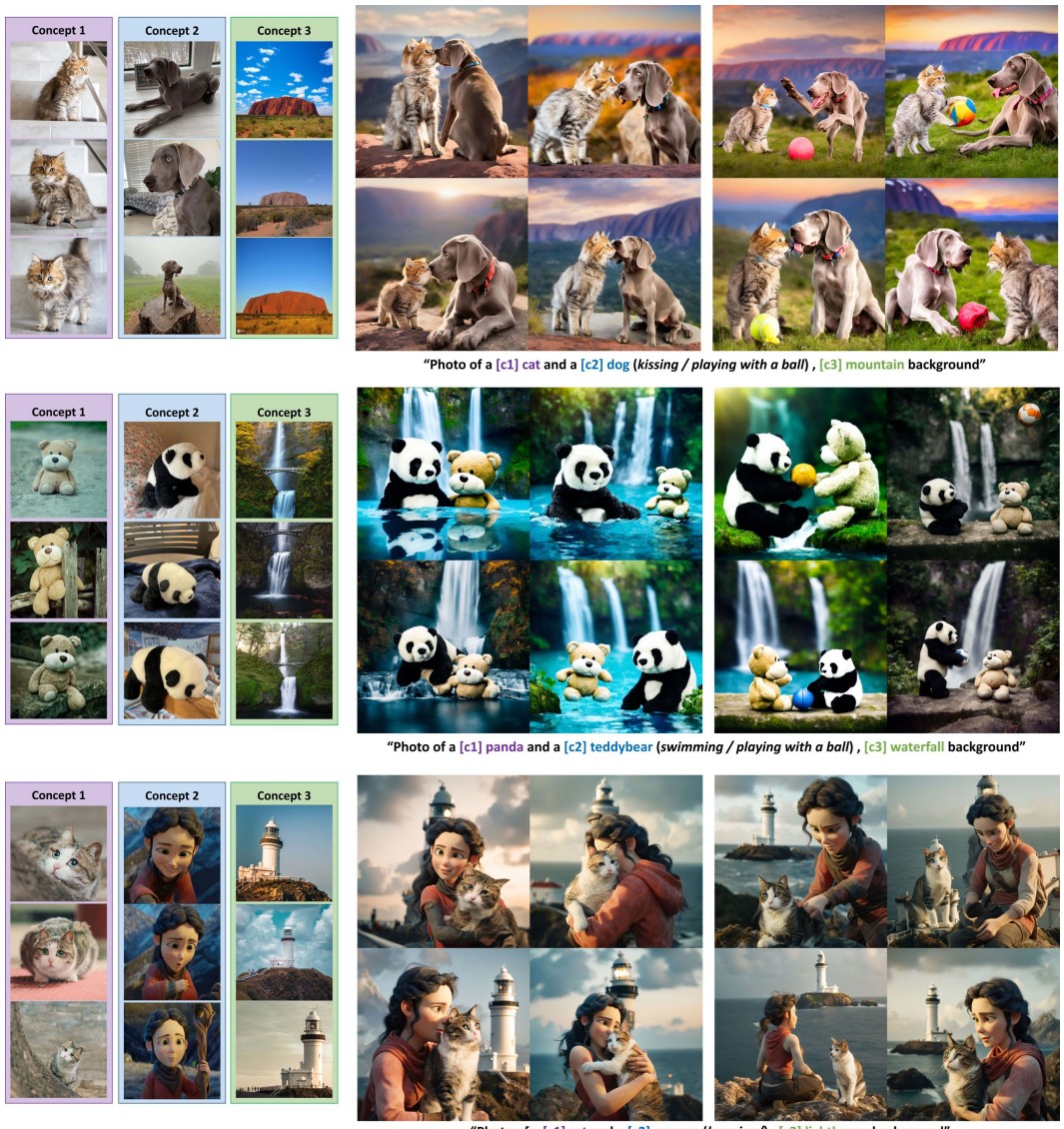

Figure 17: **Additional Results.** Our model can generate high-quality multi-concept generation results on image domains.

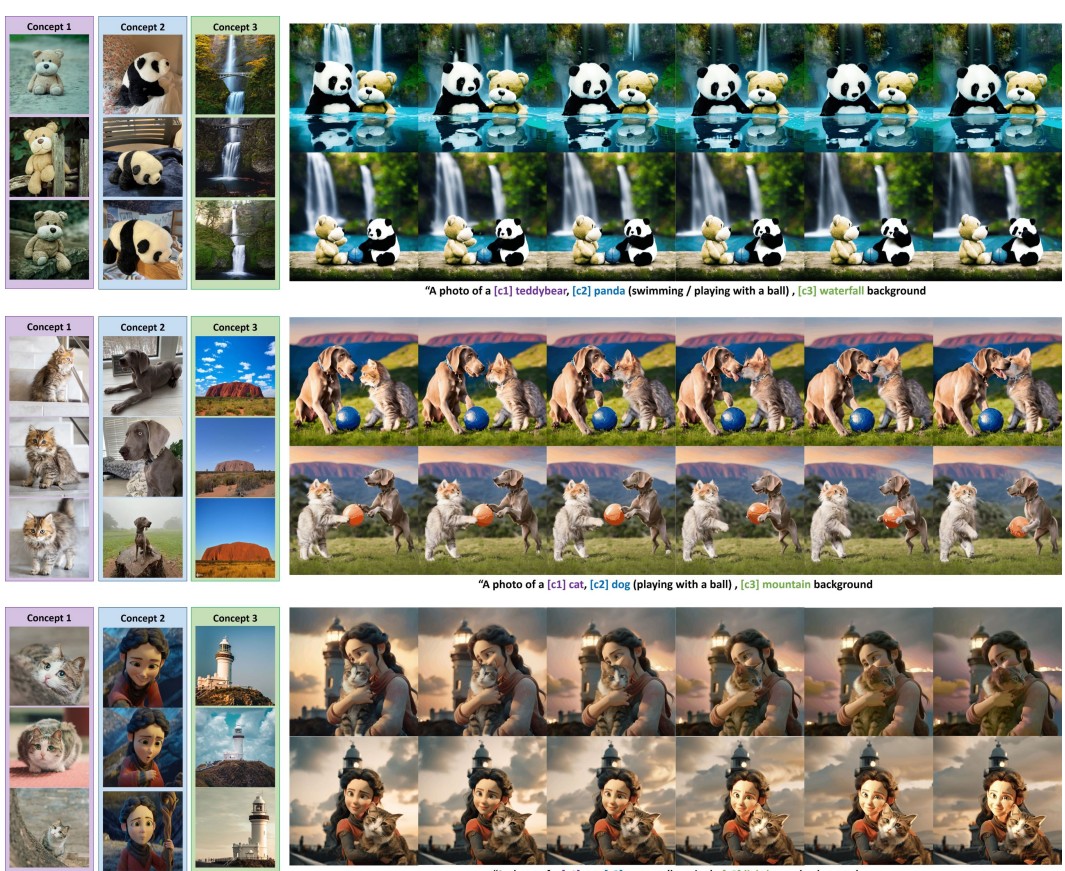

Figure 18: **Additional Results.** Our model can generate high-quality multi-concept generation results on video domains.

