# OpenReview forum: "TweedieMix: Improving Multi-Concept Fusion for Diffusion-based Image/Video Generation"
_ICLR.cc/2025/Conference — ICLR 2025 Poster_

### Official Review · Reviewer_y5WZ · 2024-10-17

**Soundness:** 3
**Presentation:** 3
**Contribution:** 2
**Rating:** 6
**Confidence:** 3

**Summary:**

The paper introduces TweedieMix, a novel method designed to customize diffusion models during the inference phase, particularly for the challenge of integrating multiple personalized concepts in image and video generation. It operates by dividing the reverse diffusion sampling process into two distinct stages: an initial stage that uses multiple object-aware sampling to ensure the inclusion of desired target objects, and a later stage that employs Tweedie’s formula to blend the appearances of custom concepts in the de-noised image space. The results indicate that TweedieMix outperforms existing methods in generating images and videos with multiple personalized concepts with higher fidelity.

**Strengths:**

1、The multi-concept generation results looks good；
2、The authors illustrate the framework details and experimental setting well.

**Weaknesses:**

1、The whole pipeline seems naive and lack of novelty.
2、Some problems exist in experimental settings.
See questions for more details.

**Questions:**

1、Making use of Text-SAM and Tweedie’s formula do not contribute to the novelty.
2、Each concept a model. I suspect the performance gain may derived from the overfitting of specific concept. And the hard mask composition seems like a naive post processing. The whole framework is more like a system which is designed under substantial ablation experiments and lack of novelty and essential understanding towards this task.
3、The extension to video, if I do not misunderstand, is a technical trick on a off-the-shell I2VGen-XL. I2VGen-XL is finetuned from ModelScopeT2V with a large dataset. While DreamVideo use ModelScopeT2V only for subject injection with a little amount of custom videos. Maybe this need further discussion and I'd like to see opinions from other reviews.

---

> ### Author Response · Authors · 2024-11-19
> **Reply to the reviewer comments**
>
> **[Q1]Using Text-SAM and Tweedie’s formula do not contribute to the novelty.**
>
> We kindly remind that combining custom concepts using masks is standard in recent multi-concept generation methods [1-4]. The key challenge lies in achieving natural object composition once masks are provided. Existing models typically combine concepts in attention layers. To address performance degradation and high computational costs, we introduced combining custom concepts in the denoised space using Tweedie’s formula.
>
> We are the first to demonstrate the CFG++ framework's suitability for multi-concept generation, enhancing fusion stability. To improve multi-concept awareness, we firstly proposed splitting the sampling timesteps: multi-concept aware sampling in initial steps and concept fusion in later steps. Additionally, we firstly introduced multi-object resampling to prevent specific concepts from disappearing.
>
> Moreover, our method processes all steps in a single sampling step, enabling faster generation. Unlike baselines, it supports region-wise sampling without attention adjustments, allowing flexible concept fusion. For instance, we seamlessly combined models fine-tuned with low-rank adaptation and attention weights fine-tuned as in custom diffusion—unachievable with baselines. These contributions are novel and represent a significant advancement.
>
> **[Q2] The performance gain may comes from overfitting to specific concepts. mask composition seems to be a naive post-processing step. The framework seems to be from extensive ablation study, lacking novelty and a fundamental understanding.**
>
> Contrary to the reviewer’s misunderstanding, we demonstrated that our performance improvements are not limited to specific concepts by validating our method on the large customconcept101 dataset. Since our baseline, Concept Weaver, was tested on a similar number of concepts, we ensured comparable experimental conditions for fairness.
>
> To further address the reviewer’s concerns, we expanded the variety of custom concepts and included new quantitative evaluation results in Table 1. We also include more comprehensive metric, DINO scores. Even with the expanded set and new metrics, ours consistently outperformed the baselines, demonstrating that the results reflect the general effectiveness of ours rather than overfitting.
>
> As a reminder, mask composition is a standard technique in recent multi-concept generation. The critical aspect lies in how naturally regions defined by masks can be synthesized, making the use of masks a non-issue.
>
> The novelty of our method lies in achieving high-quality concept fusion by combining concepts in the Tweedie-denoised space of the CFG++. We emphasize that this fusion is not limited to specific settings or concepts. To address the reviewer’s concern, Figure 10 includes experiments showing our method can combine not only custom concepts but also a wide range of styles (e.g., 3D + anime), a task unattainable with previous methods.
>
> **[Q3] The extension to video is a technical trick on a off-the-shelf I2VGen-XL. I2VGen-XL is trained on large image data while Dreamvideo use modelscope T2V with little amount of custom videos.**
>
> Thank you for your comment. We acknowledge this comparison may not be entirely fair. However, fine-tuning traditional T2V models consistently fails to generate multi-concept images. Techniques like Custom Diffusion and Textual Inversion produced unsatisfactory results, and even the latest method, DreamVideo, barely incorporated multiple custom concepts (Figure 7). Due to significant degradation, no prior research has addressed multi-concept video generation. To overcome this, we are the first to achieve it using an Image-to-Video model—an accomplishment unattainable with baselines. We emphasize that our contribution lies in addressing this new challenge.
>
> We kindly remind the reviewer that our approach is not a simple technical trick using I2VGen-XL. Directly using the I2V model often results in distortions, making it unsuitable for multi-concept video generation. To address this, we devised a novel method to extend residual layer features across frames, preserving the context of multi-concept scenarios. This simple yet effective method is a key technical novelty in our video extension, as demonstrated in Figure 8 and abstract.
>
> To show our method is not limited to I2VGen-XL, we experimented with Stable Video Diffusion(Figure 12). Applying our feature injection, we can still preserve the context information, demonstrating that our approach is not restricted to specific video models.
>
> **Reference**
>
> [1]Mix-of-Show: Decentralized Low-Rank Adaptation for Multi-Concept Customization of Diffusion Models, Neurips 2023
>
> [2]Cones 2: Customizable Image Synthesis with Multiple Subjects, Neurips 2023
>
> [3]Concept Conductor: Orchestrating Multiple Personalized Concepts in Text-to-Image Synthesis, Arxiv 2024
>
> [4]Concept Weaver: Enabling Multi-Concept Fusion in Text-to-Image Models, CVPR 2024

---

> > ### Comment · Reviewer_y5WZ · 2024-11-20
> >
> > Related to Q2:
> > I check Concept Weaver just now and I find a 'concept bank training' in the pipeline of it. I am not a expert in personalization and I have a question. Was this step "Fitting a sample-wise feature for each concept" a standard operation in recent works?

---

> > > ### Author Response · Authors · 2024-11-20
> > > **Reply to the reviewer question**
> > >
> > > Yes, most of the recent methods require sample-wise (or single-concept wise) weight fitting, and combine those weights for multi-concept generation. All of the papers that we referenced [1-4] also require sample-wise fitting process. Beyond these referenced papers, most diffusion model personalization methods require concept-wise weight tuning process.

---

> > > > ### Comment · Reviewer_y5WZ · 2024-11-20
> > > >
> > > > My mistake, I accept part of the reasons that previous work use this as a standard process. But I expect a unified method to fuse multi-concept without per-sample tuning. I will change the Rating from 5 to 6. I would like to see replies from other reviewers for further score raising.

---

> > > > > ### Author Response · Authors · 2024-11-20
> > > > >
> > > > > Thank you for raising the score and the constructive feedbacks!

---

### Official Review · Reviewer_yJkf · 2024-11-02

**Soundness:** 3
**Presentation:** 3
**Contribution:** 2
**Rating:** 6
**Confidence:** 3

**Summary:**

This paper introduces a new approach called TWEEDIEMIX for multi-concept text-to-image generation. The author introduce a tuning-free approach during the inference stage and divides the process into two main stages. The key contribution of this paper is using a resampling strategy and multi-concept fusion sampling enabling generate multiple personalized concepts with higher fidelity than existing methods.

**Strengths:**

1. The idea is easy to follow and the paper is well written.
2. The results on Custom Concept 101 dataset outperforms the previous baselines, demonstrating better generation qualities compared to the concept personalization methods.
3. The author's code has been open-sourced, which is somewhat helpful to the community.

**Weaknesses:**

1. My main concern lies in the technical contribution of this paper. It seems like an incremental work of ConceptWeaver[1] which is also a training-free method that combines multiple concepts during inference.
2. In terms of qualitative results, there doesn't seem to be a significant improvement compared to ConceptWeaver which is also can handle more than two concepts.


[1] Gihyun Kwon, Simon Jenni, Dingzeyu Li, Joon-Young Lee, Jong Chul Ye, and Fabian Caba Heilbron. Concept weaver: Enabling multi-concept fusion in text-to-image models. In Proceedings of the IEEE/CVF  (CVPR), pp. 8880–8889, June 2024.

**Questions:**

Please refer to the weaknesses above.

---

> ### Author Response · Authors · 2024-11-19
> **Reply to the reviewer comments**
>
> **[Q1] My main concern lies in the technical contribution of this paper. It seems like an incremental work of ConceptWeaver[1] which is also a training-free method that combines multiple concepts during inference.**
>
> We kindly remind the reviewer that our proposed method is different from the baseline of Concept Weaver in the following parts.
>
> (1) The key idea of concept weaver is to separate the sampling processes into multiple stages, therefore the method requires excessive time as it use all of basic image generation, image inversion with DDIM, and concept fusion stages for single output. However, our proposed method can generate multi-concept output with only single generation path, which reduces significant computational time. Specifically, the baseline of concept weaver takes about 65 seconds for single image output, but our method takes only 28 seconds for one image. (see time Appendix C)
>
> (2) For concept fusion, concept weaver uses mask guidance in cross attention layer features, which requires deliberate engineering of layer selection and requires excessive calculation. Also, as the method requires manipulation on both of the cross-attention and self attention layers, the method reduces flexibility as it cannot be adapted to different model architectures.
>
> Our proposed TweedieMix proposes to mix the concepts in denoised image space, therefore we do not require any manipulation on attention layer. It enables us more flexibility in models. For example, we can fuse the various models which are fine-tuned with Low-Rank adaptation or fine-tuned with attention weights. We kindly remind the reviewer that we have already shown that our generated outputs are mixture of various customization methods (e.g. key-value weight fine-tuned for dog , lora fine-tuned for cat)
>
> Also, our method can reduce the memory consumption as it does not require calculation on attention layers. The baseline concept weaver requires about 19GB of VRAM memory, but our method requires only 10GB of VRAM.
>
> (3) One of the key contributions of this work is the observation that simplified concept fusion is made possible by recent advancements in classifier-free guidance, specifically in the form of CFG++.
>
> Specifically, to enhance the quality of multi-concept awareness, we propose dividing the single sampling path into multi-concept-aware sampling steps and concept fusion steps. Additionally, we introduce a novel resampling method to further strengthen multiple-object generation, leveraging the inherent properties of Tweedie’s denoised space in CFG++. Furthermore, to the best of our knowledge, no other work beyond ours has successfully demonstrated multi-concept fusion in video generation.
>
> Given these innovations and the significant improvements in generation quality, we argue that our proposed method represents a substantial advancement and cannot be considered merely incremental compared to Concept Weaver.
>
> **[Q2] In terms of qualitative results, there doesn't seem to be a significant improvement compared to ConceptWeaver which is also can handle more than two concepts.**
>
> We would like to assure the reviewer that the impression of minimal differences in qualitative results may stem from our choice to use the same concepts as those in the original Concept Weaver paper for a fair comparison. While we acknowledge that Concept Weaver performs relatively well on these specific concepts, our method still demonstrates superior performance both quantitatively and qualitatively.
>
> However, to address the reviewer’s concerns, we have also included additional comparison images in Figure 11, 13,14,15. In these images, it can be clearly observed that Concept Weaver shows unwanted transformations or disappearance of custom concept objects, while our model produced significantly better visualization results than Concept Weaver.
>
> Moreover, the superiority of our results over Concept Weaver’s is more thoroughly demonstrated in the quantitative results and user study. To further address the reviewer’s concerns, we conducted additional detailed comparison experiments by adding the DINO score and expanding the pool of concept images in our Table 1.

---

> > ### Comment · Reviewer_yJkf · 2024-11-27
> >
> > I have read the rebuttal carefully and would like to thank the authors. I greatly appreciate their efforts in addressing some of my concerns regarding the technical contribution. As a result, I would like to raise my score.

---

> > > ### Author Response · Authors · 2024-11-27
> > >
> > > Thank you for the constructive comments and raising the score!

---

> ### Author Response · Authors · 2024-11-23
> **Nearing  the end of the discussion period**
>
> Dear Reviewer yJkf,
>
> As the deadline for the Reviewer-Author discussion phase is fast approaching, we respectfully ask whether we have addressed your questions and concerns adequately.
>
> Sincerely,
>
> The Authors.

---

### Official Review · Reviewer_JzyV · 2024-11-03

**Soundness:** 3
**Presentation:** 3
**Contribution:** 3
**Rating:** 8
**Confidence:** 3

**Summary:**

This paper proposes a method to fuse multiple concepts for personalized T2I diffusion-based models that are generalizable to both image and video generation. The method mainly involves two parts: (1) concept-aware sampling that coarsely localizes the concepts in the latent space, and (2) multi-concept fusion that generates the personalized concepts based on the extracted regional masks. The qualitative and quantitative results both demonstrate the effectiveness of the proposed method.

**Strengths:**

+ The proposed method is reasonable and intuitive.
+ The generated results of multi-concept fusion are impressive.
+ The model is properly extended from image generation to video generation.
+ From the quantitative results, the proposed method outperforms previous models.

**Weaknesses:**

- $t_con$ and $P$ are two important hyperparameters. Can the authors provide generated results with varying $t_con$ and $P$, while keeping all other settings fixed?
- DINO [1] can extract subtle visual features that are not specific to any particular category. The metric of DINO image similarity should also be reported.
- The evaluation is conducted on combinations of more than three concepts; however, the paper does not present any results of combinations involving more than three concepts.
- It is recommended to compare the sampling time of the proposed method with that of other models.
- Missing reference on unsupervised multi-concept extraction [2].

[1] Caron etal. Emerging Properties in Self-Supervised Vision Transformers. ICCV 2021.
[2] Hao et al. ConceptExpress: Harnessing Diffusion Models for Single-image Unsupervised Concept Extraction. ECCV 2024.

**Questions:**

- Please see the weaknesses.

---

> ### Author Response · Authors · 2024-11-19
> **Reply to the reviewer comments**
>
> **[Q1] $t_{con}$ and $P$ are two important hyperparameters. Can the authors provide generated results with varying $t_{con}$ and $P$, while keeping all other settings fixed?**
>
> Thank you for your suggestions, We included the results with varying the parameters in Appendix B and Figure 9.
>
> **[Q2]DINO [1] can extract subtle visual features that are not specific to any particular category. The metric of DINO image similarity should also be reported.**
>
> Thank you for your constructive comment. We have included DINO image similarity score In Table 1, and included explanation on section 5.1 and Appendix A.
>
> **[Q3]The evaluation is conducted on combinations of more than three concepts; however, the paper does not present any results of combinations involving more than three concepts.**
>
> For quantitative score calculation, we have already included the experiments on four concepts. To address the reviewer’s concern, we have included qualitative comparison results in Figure 11.
>
> **[Q4]It is recommended to compare the sampling time of the proposed method with that of other methods.**
>
> Thank you for your feedback. We have included the time efficiency comparison in Appendix C. Our proposed method takes about 28 seconds, while our main baseline method of concept weaver takes about 65 seconds.
>
> **[Q5] Missing reference on unsupervised multi-concept extraction [2]**
>
> Thank you for your recommendation. We included the reference in the related work part.

---

> ### Author Response · Authors · 2024-11-23
> **Nearing the end of the discussion period**
>
> Dear Reviewer JzyV,
>
> As the deadline for the Reviewer-Author discussion phase is fast approaching, we respectfully ask whether we have addressed your questions and concerns adequately.
>
> Sincerely,
>
> The Authors.

---

> ### Comment · Reviewer_JzyV · 2024-11-23
>
> Thank the authors for their feedback, which has addressed my concerns raised in the initial review. I have no further questions.

---

> > ### Author Response · Authors · 2024-11-23
> >
> > Thank you for the constructive comments and raising the score!

---

### Author Response · Authors · 2024-11-19
**General Reply**

We sincerely thank all reviewers for the constructive feedback. Per all the reviewers' comments, we revised the manuscript. Here is the summary of changes to our paper.

●	We included DINO image similarity score for further comparison.

●	We included new quantitative scores which are calculated using extended concept dataset (15 concepts -> 32 concepts)

●	We included additional experiment with varying the hyperparameter t_cond and P (Appendix B, Figure 9)

●	We included inference time comparison between the baselines. (Appendix C)

●	We included qualitative results on combining 4 different concepts (Fig 11, Appendix E)

●	We included more qualitative comparison between Concept Weaver and ours in Figure 11,13,14,15.

●	To evaluate the versatility of our proposed method, we included qualitative results of multiple style composition in Figure 10 and Appendix D.

●	We included missing references in related work.

●	We included video generation results on Stable Video Diffusion in Figure 12.

●	We corrected some typos.

---

### Meta-Review · Area_Chair_2kBd · 2024-12-19

**Metareview:**

This paper introduces a novel method, TweedieMix, for personalized multi-concept text-to-image and text-to-video generation using diffusion-based models. The method mainly has two key stages: concept-aware sampling for localization in latent space, and fusing appearances of personalized concepts. The results demonstrate convincing performance in combining multiple concepts, outperforming prior models in fidelity and quality on image and video generations. There are some concerns about limited novelty and incremental contributions compared to prior work like ConceptWeaver. Given the method’s practicality and effectiveness from quantitative and qualitative results, the method has potential value to the research community. The authors are encouraged to address concerns about experimental settings, novelty, and additional evaluations in their revisions.

**Additional Comments On Reviewer Discussion:**

The discussions are straight forward, since the first round reviews are mostly positive. The authors have been actively providing details such as differences against existing work, and most of the questions have been resolved.

---

### Decision · Program_Chairs · 2025-01-22

Accept (Poster)